# Crop Loss Evaluation Using Digital Surface Models from Unmanned Aerial Vehicles Data

**Virginia E. Garcia Millan** [1,*] **, Cassidy Rankine** [2] **and G. Arturo Sanchez-Azofeifa** [3]

[1] Department of Physical Geography and Ecosystem Science, Lund University, Sölvegatan 12,
    223 62 Lund, Sweden
[2] Skymatics, Calgary, AB T2P3C5, Canada
[3] Department of Earth and Atmospheric Sciences, University of Alberta, Edmonton, AB T6G 2E7, Canada
[*] Correspondence: virginia.garcia@nateko.lu.se

**Abstract:** Precision agriculture and Unmanned Aerial Vehicles (UAV) are revolutionizing agriculture management methods. Remote sensing data, image analysis and Digital Surface Models derived from Structure from Motion and Multi-View Stereopsis offer new and fast methods to detect the needs of crops, greatly improving crops efficiency. In this study, we present a tool to detect and estimate crop damage after a disturbance (i.e., weather event, wildlife attacks or fires). The types of damage that are addressed in this study affect crop structure (i.e., plants are bent or gone), in the shape of depressions in the crop canopy. The aim of this study was to evaluate the performance of four unsupervised methods based on terrain analyses, for the detection of damaged crops in UAV 3D models: slope detection, variance analysis, geomorphology classification and cloth simulation filter. A full workflow was designed and described in this article that involves the postprocessing of the raw results from the terrain analyses, for a refinement in the detection of damages. Our results show that all four methods performed similarly well after postprocessing—reaching an accuracy above to 90%—in the detection of severe crop damage, without the need of training data. The results of this study suggest that the used methods are effective and independent of the crop type, crop damage and growth stage. However, only severe damages were detected with this workflow. Other factors such as data volume, processing time, number of processing steps and spatial distribution of targets and errors are discussed in this article for the selection of the most appropriate method. Among the four tested methods, slope analysis involves less processing steps, generates the smallest data volume, is the fastest of methods and resulted in best spatial distribution of matches. Thus, it was selected as the most efficient method for crop damage detection.

**Keywords:** Unmanned Aerial Vehicles; structure from motion; precision agriculture; digital surface models; crop loss estimation

## 1. Introduction

At present, around an 11% of the global land surface (13.4 billion ha) is used in crop production (arable land and land under permanent crops) [1]. According to data of the World Fact Book [2], in 2015 agriculture accounted for 5.9% of the gross domestic product (GDP) worldwide. However, even though other economic sectors are more profitable for the national economies, the entire world population depends on agriculture and farming. This sector is under a high risk of investment given the unpredictable factors that can affect the yield, such as weather conditions, natural disturbances and market fluctuations. Furthermore, growing season coincides with the peak of disturbances: fires, summer storms, breeding of herbivores, flourishing of plagues and diseases, etc. [3–5].

Under many climate change scenarios, many authors point out that human systems developed during the Holocene (such as agriculture) are under risk of disappearing or at least be compromised in the Anthropocene era that we have entered, and it is expected that this situation might become worse in the coming years [6–8]. While the total world precipitation has not significantly changed in the last decade, the distribution of rainfall across the year and space has changed, implying relevant effects in biomes and agriculture lands [9].

Farmers and agronomists are aware of the risk of their business, with an eye on possible changes on climate. Thus, in the literature we found many studies related to alternative management practices to face climate change [10,11]. Some effects, unfortunately, cannot be prevented, such as floods, storms and fires. For this reason, an accurate evaluation and prediction of damages is of major importance for farmers and stakeholders in the agriculture business, including insurance companies. Many farmers and agriculture corporations invest in insurance in case they lose part or the totality of the yield. Traditional crop-loss assessment methods are slow, labour intensive, biased by restricted field access, and subjective from adjuster to adjuster. The subjectivity of damage estimation leads in many cases to disagreements between farmers and adjusters, in terms of how many hectares are insurable. Both crop insurers and farmers are looking for ways to improve resource management and claims reporting. The current paper proposes more objective and accurate methods to calculate crop damages by means of Unmanned Aerial Vehicles (UAV) remote sensing.

Precision agriculture is a farming management concept based on observing, measuring and responding to inter and intra-field variability in crops using remote sensing tools, such as satellites, planes or UAVs [12–15]. A review on precision agriculture companies reveals that they focus on very-high resolution sensors [16–18], multispectral cameras [19–21], hyperspectral sensors [22–24], thermal sensing [25] and wireless sensor networks [25–28], to measure plant and soil properties such as plant nutrient and water content and soil humidity. At the beginning, precision agriculture made use of light-weight remotely controlled airplanes, but in recent years, UAVs are being more and more used in this field because of their versatility [18]. Some authors have developed remote sensing methods for the evaluation of crop damage, such as the effects of droughts [29], the detection of fungal infection [30], crop losses in floods [31], and detection of crops affected by fires [32]. Nevertheless, to our knowledge, there are not many precision agriculture algorithms that account for digital canopy models of plants and damage quantification. Moreover, the existing literature only shows case studies of one or few crop types and/or damage types.

In the last decade, low-cost UAV technology has been significantly developed, encouraging the use of UAVs for commercial applications, such as precision agriculture, oil and gas, urban planning, etc., [18,33]. Now it is possible to find low-cost UAVs, cameras and software in the market that collect high-resolution georeferenced imagery, which can be later used to generate Digital Surface Models (DSM) and 3D point clouds by means of Structure from Motion (SfM) and Multi-View Stereopsis (MVS) [34–37]. Specifically, low-cost light UAVs dotted with RGB cameras are the most preferred Unmanned Aerial System (UAS) by many users, farmers among others. Whereas RGB cameras are limited by the development of spectral products, beyond scouting and monitoring, DSMs avoid artefacts that appear often in the UAV imagery, such as shadows projected from objects in the surface or clouds, light differences during the flight time or different colours caused by soil moisture or wet vegetation [38–40]. These challenges can be addressed with deep learning, but they need a large labelled database to feed the algorithm [41]. While UAV-derived DSM from commercial UAVs have been largely used for small scale terrain mapping and 3D modelling of urban areas [42–44], they have been also proved useful for agricultural applications [45–50].

As such, this study explored four terrain analysis tools, based on point cloud data or DSM, for the delineation and quantification, in hectares, of damages in field crops due to insurable causes, such as weather events and wildlife attacks. For this study, we focused on events that cause severe physical damages in the plant structure, generating abrupt depressions in the surface of the canopy, from which the plant rarely recovers. This type of damage is the most distressing for farmers and appointed by

adjusters for monetary compensation. By the digital 3D reconstruction of the crop canopy, it is possible to detect differences in crop height or dramatic changes in slope that are related to damages. The accuracy of four different analysis methods are here evaluated and a workflow is proposed to deliver a digital product that can be translated into the number of hectares of damaged crop. The proposed workflow can be easily applied to any type of UAS equipped with snapshot cameras or video recorders, which later produce SfM and MVS-based DSMs. Our contribution to the field of precision agriculture is the assembling of a tool from existing methods that has been proven successful as a generic crop damage estimator, independent of the crop type, damage type or growth stage.

## 2. Materials and Methods

### 2.1. Dataset Collection

Six sites in Europe and America corresponding to different crop types, damage types and growth stages were selected for this study (Figure 1, Table 1). These sites present different climates and soil conditions, and are covered by dry croplands (wheat and barley) and irrigated croplands (corn). Our dataset comprehends the point clouds, DSMs and orthomosaics from six field crops that were damaged and later imaged with an UAS. Volunteer farmers that suffered damages in their croplands donated the data. A code was assigned to each dataset to keep the confidentiality of the participants (Table 1). One goal of this study is to evaluate the potential of unsupervised remote sensing methods to deal with generic DSMs data. Therefore, no detailed information about the crops and the events that generated the damage was requested to the farmers. Also, no information about the UAV or camera models, or flight conditions was available.

**Table 1.** Study sites described by crop type, crop damage and growth stage.

| Dataset Code * | Crop Species | Damage Type | Growth Stage ** | Area (ha) | Site |
|---|---|---|---|---|---|
| B1L | Barley | Logging | 73. Milking | 79.4 | Canada |
| W1W | Wheat | Wind | 85. Ripening | 35.4 | Hungary |
| W2W | Wheat | Wind | 89. Late ripening | 3.9 | UK |
| C1M | Corn | Man-made | 69. Flowering | 4.1 | Brazil |
| C2A | Corn | Wildlife | 31. Jointing | 7.1 | Hungary |
| C3W | Corn | Wind | 49. Booting | 34.7 | Chicago, US |

* Code naming: Crop type—sample #—Damage Type; Crop Type: B: barley, C: corn, W: wheat; Damage type: L: Logging, M: man-made, A: animals, W: wind. ** According to BBHC growth scale.

The data were selected to represent different situations that might affect the results of the terrain analysis: crop type, damage type and growth stage. Plant structure, and particularly crop canopy, are related to the crop type (species) and growth stage; the height of the plant, density and distribution, leaf clumping, texture of the canopy, etc., affect the surface of the canopy, and therefore, the DSM. On the other hand, damage type and intensity define the geometry and size of the damage, which is expressed as depressions in the crop canopy. This is also reflected in the DSM. To cover different scenarios, crops with significant different height and vertical geometry were selected: corn vs. barley and wheat. In the same fashion, different damage types were selected to have a variety of situations: logging, wildlife attacks, windstorms and man-made damages. To add heterogeneity to the dataset, each crop in the dataset was in a different growth stage (Table 1).

DroneDeploy [51] (DroneDeploy, San Francisco, CA, US) was used as the platform where the participants of the study uploaded their datasets. The data was nominally generated at 5 cm pixel size, but this fine resolution increases and hinders greatly the computational time. Therefore, the orthomosaics and DSMs were downloaded with a resolution of 10 and 50 cm pixel resolution. In addition, for some of the tested methods, a larger pixel size is recommended to identify the canopy depressions as an entity [52]. The Digital Surface Models were processed and generated by DroneDeploy, which use an algorithm based on SfM [53–59].

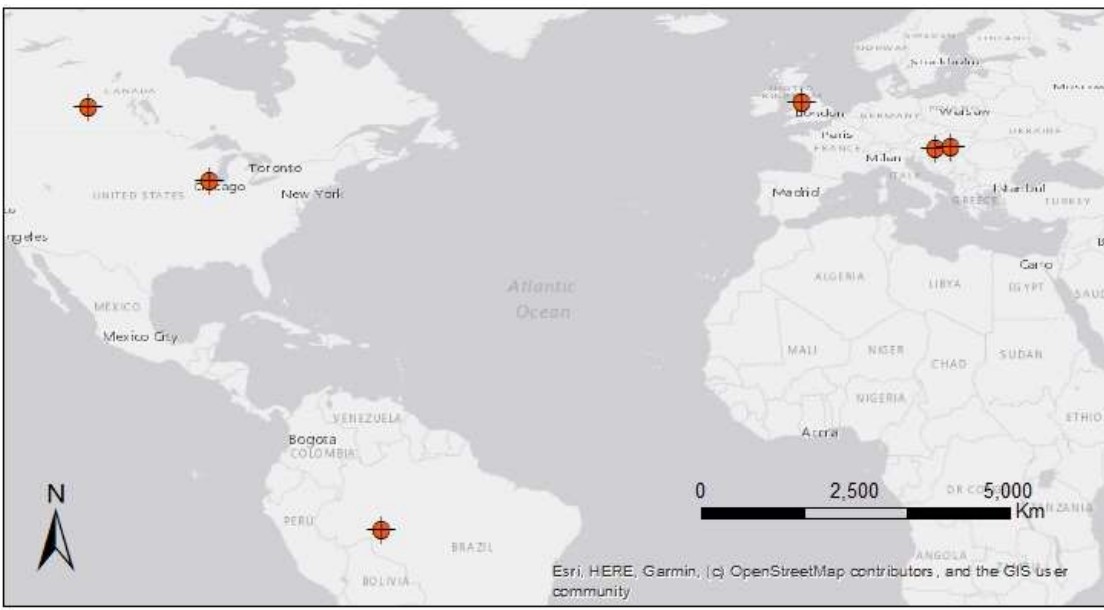

**Figure 1.** Location of study sites.

## 2.2. Workflow to Detect Crop Damages Using UAV-DSM

Certain events, such as windstorms and wildlife attacks, affect the structure of plants in the form of depressions on the crop canopy. These damages cause a dramatic change of elevation between healthy and damaged vegetation, and are visible in a DSM. This motivated the selection of methods that can detect changes in elevation directly from the data. Surface morphology has been defined using DSMs in other disciplines such as topography, hydrology, forestry or urban planning [60–63]. In this study, we took advantage of existing algorithms for the delineation of crop damages.

However, there can be other elements in the DSM that cause discontinuities or depressions in the crop canopy, such as machinery tracks or water ponds, which can be wrongly classified as damage in terrain analysis. In order to distinguish crop damages from other elements of the landscape, additional processing steps were implemented before and after the terrain analysis, based on spatial characteristics of the different depression types. Overall, we are presenting a workflow in this article, which includes (1) the preprocessing of the input DSMs, (2) terrain analysis and (3) postprocessing of the outcome results (Figure 2).

### 2.2.1. Terrain Analysis Methods

In this study, four terrain analysis algorithms were tested: slope detection [64], variance analysis [65], geomorphology classification [52,66] and cloth simulation filter [18] (Table 2).

Slope detection and Variance were developed to detect changes in topographic elevation. Geomorphology classification was developed in the field of hydrogeology, to locate areas where water would run off or accumulate in a given landscape. Cloth simulation filter was designed to separate digital surface and ground points from a LiDAR dataset of urban or forested areas.

From an analytical point of view, Geomorphology classification [66,67], Slope [64] and Variance [65] compare each pixel of the scene with the neighbouring pixels to discern the relative elevation in respect to their surroundings. These are known as "kernel-based" methods. On the contrary, Cloth Simulation Filter (CSF) [18] considers the relative position of points and surfaces in reference to all the observed area.

The results of Slope and Variance are rasters in floating data format. In this study, low values correspond to flat or invariant areas (standing vegetation or bent vegetation); high values correspond to areas that present a significant change in elevation (edges of damage).

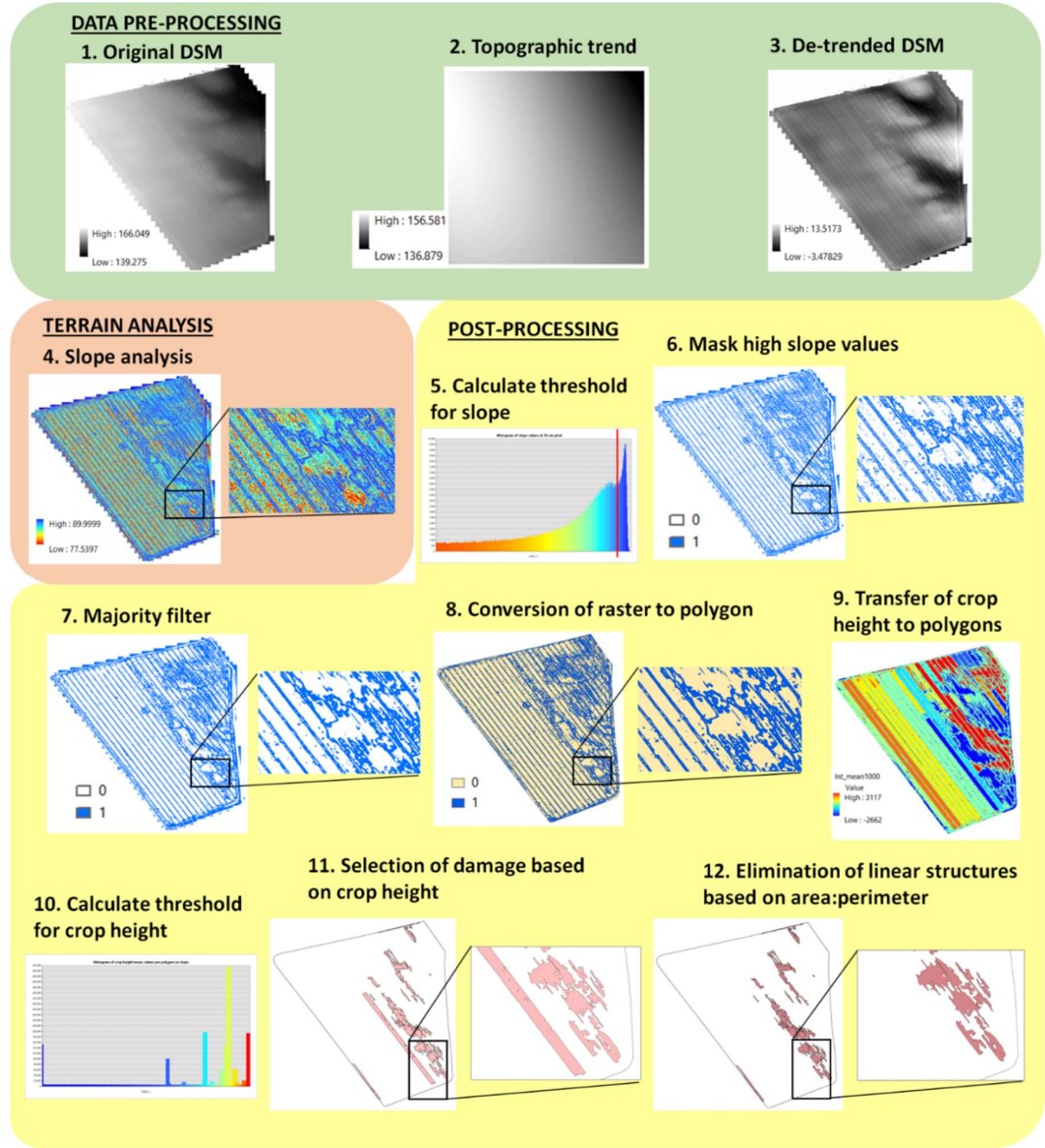

**Figure 2.** Full crop damage detection process, including pre and postprocessing workflows and 3D analysis. Slope analysis on study site "W1W" as example. See Table 1 for description of the site.

**Table 2.** General description of terrain analysis methods in this study.

|  | **Slope Detection** | **Variance Analysis** | **Cloth Simulation** | **Geomorphology Classification** |
|---|---|---|---|---|
| Software | ArcMap | ENVI | Cloud Compare | GRASS |
| File format | Raster | Raster | Point cloud | Raster |
| Statistics calculation | Kernel-based | Kernel-based | Point cloud | Kernel-based |
| Topography | Any | Any | Detrended | Any |
| Output | Edge of damage | Edge of damage | Depression area | Depression area |
| Output format | Raster | Raster | Points | Raster |

Slope Detection

Slope was calculated in ArcGIS 10.2. (ESRI, Redlands, CA, US) using the DSMs in raster format. The calculation of slope is based on a trigonometric rule (1) [64]. For clarity in the equations, please follow the naming of pixels of Figure 3:

$$Slope = tan \ \theta \left( \sqrt{([dz/dx]2 + [dz/dy]2)} \right) \tag{1}$$

where *tan* is the tangent of the angle *θ*; *x* and *y* are dimensions in the horizontal plane and *z* is the height. In this case, slope is calculated in a 3D model; therefore, both the relationship between *x* vs. *z* and *y* vs. *z* are calculated. Considering that we would like to calculate the slope of the central pixel *e* in a kernel window of 3 × 3 pixels, the change of direction rate is calculated in the *x* axis as:

$$[dz/dx] = ((c + 2f + i) − (a + 2d + g)/(8 × x\_cellsize) \tag{2}$$

where *c*, *f* and *i* are the pixels on the right column to e while a, d and g are the pixels on the left column. In the same way, the change of direction rate is calculated in the *y* axis as:

$$[dz/dy] = ((g + 2h + i) − (a + 2b + c)/(8 × y\_cellsize) \tag{3}$$

where *g*, *h* and *i* are the pixels on the top row to *e* while *a*, *b* and *c* are the pixels on the bottom row. As a summary, taking the rate of change in the x and y direction, the slope for the center cell *e* is calculated using:

$$slope = \sqrt{\left( \left[ \frac{dz}{dx} \right]^2 + \left[ \frac{dz}{dy} \right]^2 \right)} \tag{4}$$

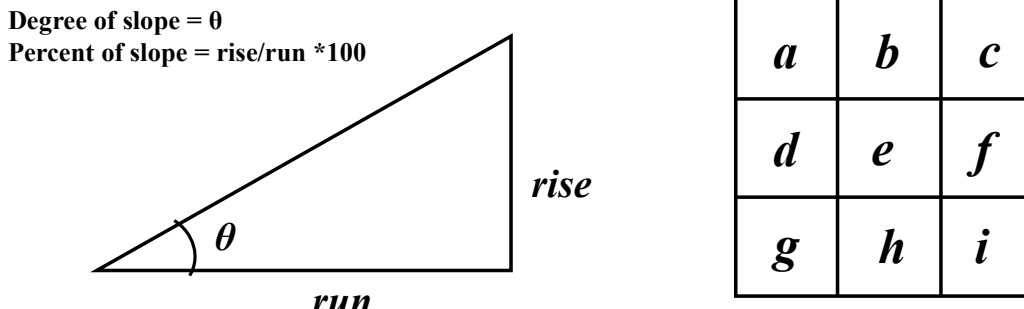

**Figure 3.** Trigonometric fundamentals of slope calculation. Modified from [68].

Variance Analysis

　　Variance is a measure of the dispersion of values around the mean. In the case of our study, the variance is calculated on the DSMs, where the values of pixels refer to crop height, and compares pixels within a determined kernel size. First-order metrics operate on the counts (occurrences) of the different digital number (DN) values within a kernel. For the aim of comparison, the same 9 × 9 kernel size was defined for the six tested DSMs.

　　Variance was calculated in ENVI 5.0. [67] (Exelis Visual Information Solutions, Boulder, CO, US). For each pixel in the image, ENVI obtains the normalised histogram of the values in the kernel that determines the frequency of occurrence of the quantised values within the kernel. Then, it computes the first-order occurrence metrics to calculate the variance [65]:

$$Variance = \sum_{i=0}^{Ng-1} (i − M)^2 \ x \ P(i) \tag{5}$$

where *i* is the pixel value, *M* is the mean value of all pixels in the kernel, *P(i)* is the probability of each pixel value, and $N_g$ is the number of distinct grey levels in the quantised image [65]. *P(i)*, or probabilities, are the normalised occurrence values, obtained by dividing the pixel values by the number of values in the kernel [65].

Geomorphological Tools

Geomorphology classification is a module (r.geomorphon) in GRASS 7.8.2. [68] (OsGeo, Beaverton, OR, US) designed for classifying landforms from DSMs in raster format [52,66]. Originally, it was designed for satellite-derived data. The Geomorphon tool analyzes the relationship of a given point regarding its surroundings. Eight orthogonal lines are traced having the observed point in the center. The intersection of the DSM with the lines marks the reference points to determine if the surroundings are higher, lower or at the same level than the observed point.

Form recognition depends on two parameters; search radius and flatness threshold. Small values of radius identifies landforms of a size smaller than the search radius, while landforms having larger sizes are broken down into components. Larger values of search radius allow for simultaneous identification of landforms on variety of sizes in expense of recognizing smaller, second-orders forms. Flatness threshold is the z-values surface that separates the point cloud into ground points and not-ground points. There are two additional parameters: skip radius and flatness distance. Skip radius is used to eliminate impact of small irregularities. On the contrary, flatness distance eliminates the impact of very high distance (in meters) of search radius, which may not detect elevation difference if this is at very far distance. Flatness distance is important especially with low resolution DSMs [52,66].

Cloth Simulation Filter (CSF)

Cloth Simulation Filter (CSF) is a plug-in into CloudCompare [18] to extract ground points in discrete return LiDAR point clouds. The parameters that define the cloth are:

1. Cloth resolution. It refers to the grid size (the unit is the same as the unit of point clouds) of cloth which is used to cover the terrain ($xy$ dimensions). The bigger the cloth resolution is set, the coarser the Digital Terrain Model (DTM) achieved.
2. Classification threshold. It is the distance between points in the $z$ dimension. It refers to a threshold (the unit is the same as the unit of point clouds) to classify the point clouds into ground and nonground parts based on the distances between points and the simulated terrain.

The tool simulates laying a cloth on an inverted Digital Elevation Model (DEM), connecting the points in the cloud that define the edges of elevated objects. The cloth represents the ground points of a DEM, and the rest the objects with certain elevation. The purpose in our study is to use it on DSM of the crops to separate the surface of the healthy crops from the depressions on the canopy, which correspond to damaged crops. The rationale behind it is the opposite of the common use of CFS: in our case, we are not separating elevated objects from ground, but crop canopy surface from depressions.

Table 3 summarises the parameters that were chosen for the different tested terrain analyses. Except for CSF's cloth resolution, all parameters were the same for all study cases to ensure consistency: a kernel of $9 \times 9$ pixels was used for the calculation of the Variance; a classification threshold of 0.2 was used for CSF; while for the Geomorphology classification, an outer search radius of 60 and a flatness threshold of 1 were set. Unfortunately, it was not possible to find a logic for cloth resolution that responds to the data characteristics (i.e., pixel resolution, topography, etc.); therefore, the thresholds were selected manually after testing different values and choosing the best result visually. In addition to the parameters shown in Table 3, a threshold to separate damage and no-damage in slope and variance was calculated, using a logistic function [69].

2.2.2. Data Preprocessing

The preprocessing step of the workflow involved (1) the detrending of the data and (2) the projection of the point clouds into Universal Traverse Mercator (UTM), and (3) the conversion of the DSM into a point cloud. Steps (2) and (3) were only required for the CSF method.

The detrending of the digital elevation data (DSM or point cloud) aimed to remove the topography trend and keep only the height of the objects in the surface—crops, in this case. The specific term in precision agriculture for the product of this process is Crop Surface Model (CSM), which refers

to the 3D representation of the crop heights, alone. However, for simplicity, we will keep the term "DSM" in the core of the text. There are two main reasons for detrending the data. On the one hand, the Cloth Simulation Filter (CSF) is sensitive to topography. Since crop depressions are less evident than terrain topography, a previous detrending of the DSMs is necessary to locate crop depressions in nonflat terrains. On the other hand, despite Slope, Variance and Geomorphology classification methods are independent of the general topography of the terrain (since the analysis is made at a kernel level), detrended DSMs were also necessary later in the postprocessing step. Therefore, all DSMs were detrended at the beginning of the workflow [63].

**Table 3.** Parameters defined for each of the 3D analysis methods. All units are "pixels".

|  | Variance | Cloth Simulation Filter (CSF) | | Geomorphology Classification | |
| --- | --- | --- | --- | --- | --- |
|  | Kernel Size | Classification Threshold | Cloth Resolution | Outer Search Radius | Flatness Threshold |
| B1L | $9 \times 9$ | 0.2 | 3.0 | 60 | 1 |
| W1W | $9 \times 9$ | 0.2 | 2.0 | 60 | 1 |
| W2W | $9 \times 9$ | 0.2 | 2.0 | 60 | 1 |
| C1M | $9 \times 9$ | 0.2 | 4.0 | 60 | 1 |
| C2A | $9 \times 9$ | 0.2 | 1.0 | 60 | 1 |
| C3W | $9 \times 9$ | 0.2 | 2.0 | 60 | 1 |

The detrending process requires a first step of calculating the trend of the terrain, which is calculated on a grid of points that contain the elevation values. The DSMs were transformed into a point structure, transferring the elevation value to a point in the center of each pixel of the raster. The trend was calculated in ArcMap [69] with a linear regression (polynomial order 6), which removed the topography without eroding the elements of the surface. The last step of the preprocessing involved the subtraction of the trend from the original DSM (Figure 4).

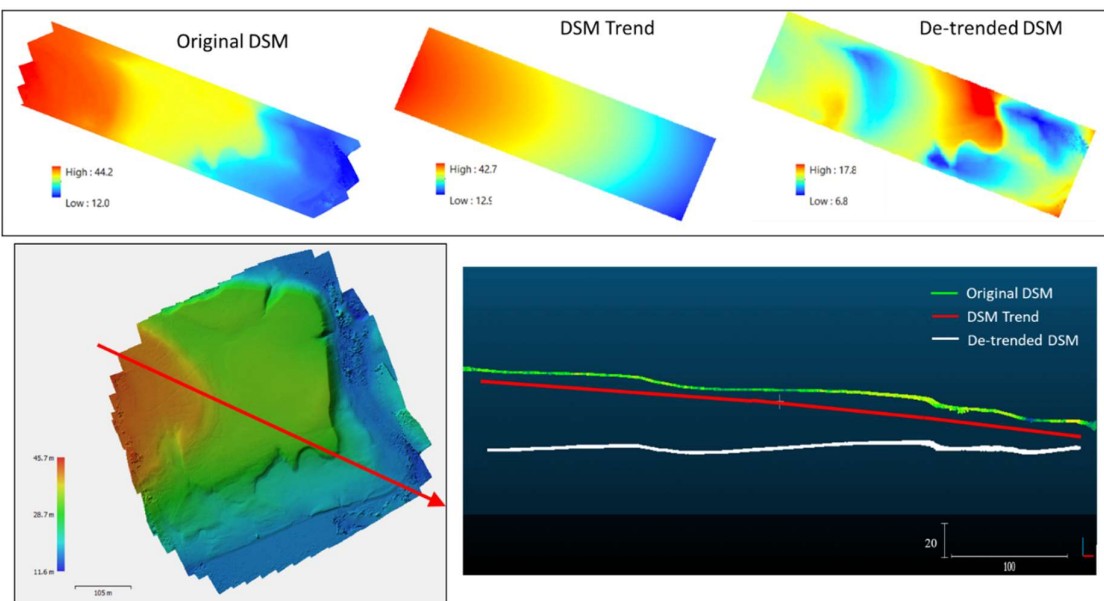

**Figure 4.** Calculation of topography trend using linear polynomial factor 6 and detrending of Digital Surface Models (DSM).

CloudCompare requires the input data to be projected in Universal Transverse Mercator (UTM). Since detrended point cloud was needed for CSF, we derived the points from a CSM that was projected into UTM, in ArcMap.

### 2.2.3. Data Postprocessing

The postprocessing step was meant to (1) convert data formats for the next steps of the workflow, (2) select the data from the outcomes of the analyses that correspond to damaged vegetation (3) reduce errors in the results of the terrain analysis and (4) calculate the damages areas in hectares (Figure 5).

To start, the tested terrain analysis methods do not deliver "damages", directly. Variance and Slope detect the edges of the damage but not the damaged area. Geomorphology classification delivers a map of geomorphologies, from which only classes that represent depressions are of interest. Only in the case of CSF, the output of the analysis is directly damaged areas, but the output is not in the appropriate format.

Afterwards, the refined results of the methods were transformed into a polygon shape, to be able to calculate the damaged areas in the crops under study.

For Slope, values close to vertical, 90 degrees, depict the edge of the damage. For Variance, high variability means a change of elevation, therefore the edge of the damage. Once the edge of the damage was delineated by a threshold, it was required that we label which areas are within the edge (damage) and out of the edge (no damage). This classification was made based on the elevation of the crops, calculated from the CSM.

First, it was required to select the threshold value that represents the edge of the damage. We used a logistic function over the histogram of values defined as [70]:

$$d(t) = 1/(1+e^{-t}) \qquad (6)$$

where *d(t)* is the logistic function and *t* is a parameter to be fitted to our data.

The logistic model describes a sigmoid curve separating the values into two groups. The inflexion point of the curve projected on the x-axis is the threshold that separates the edges of damage from the rest of values (standing and damaged vegetation). The result of this operation is a boolean raster where 1 refers to the edge of the damage and 0 to anything else (crop canopy and damaged crops).

The boolean rasters were converted into a polygon vector, and the average height of each polygon was calculated by crossing the polygons with the CSM. This allowed for interpreting which areas were damage (low crop height) or standing vegetation (high crop height).

Again, the logistic function was used to separate polygons with high and low crop height, by the evaluation of histogram of heights. Therefore, the threshold for damage edge and the damaged areas were not selected manually by the user, but the decision came from the data itself.

In the case of the Geomorphology classification, classes 9—valley and 10—depression represent crop damages. For this reason, these two classes were selected from the geomorphology classification map.

CSF separates ground points, from the whole point cloud. In this study, those points are depressions in the canopy surface (crop damage). To calculate the area that those ground points cover, they were transformed into a surface, in the form of raster or polygon.

As expected, the terrain method selected all canopy depressions that were present in the data. To keep only those that correspond to crop damage areas, the postprocessing aimed to identify and remove nondamage areas, such as machinery tracks and ponds.

Several spatial analyses were used to remove errors and enhance the accuracy crop damage detection. For instance, sparse pixels were removed using a majority filter, using a kernel of up to $8 \times 8$ pixels and a replacement threshold to half. In addition, a ratio area: perimeter was calculated to identify and remove linear structures, which generally corresponded to machinery tracks or field edges. An area: perimeter ratio close to 1 corresponds to rounded shape-forms, while long linear structures present low area: perimeter values because of long perimeters. Also, area: perimeter values under one represent polygons with a very irregular shape. By selecting polygons with an area: perimeter between 1 and 5, damages from machinery tracks were filtered.

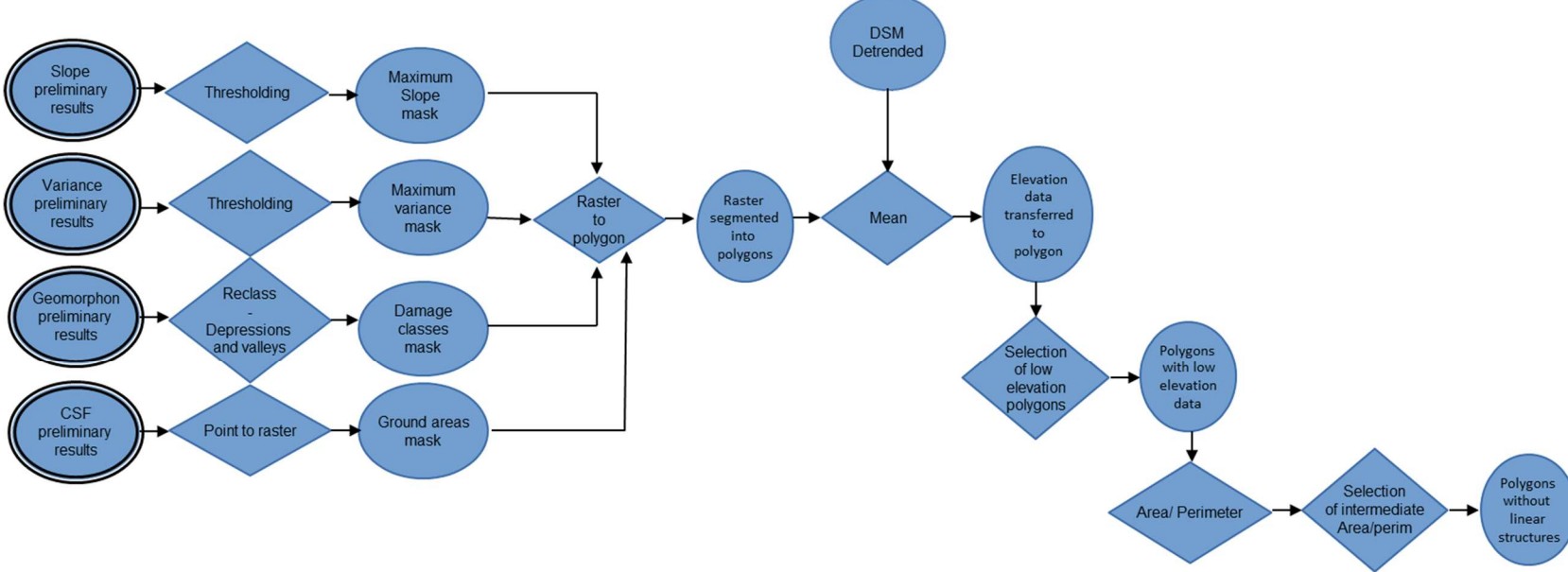

**Figure 5.** Postprocessing workflow. Circles represent data, diamonds represent processes.

### 2.3. Validation of Data

A validation workflow was designed to generate independent data that can be used to calculate the overall accuracy of the tested methods, as well as the user and producer's accuracy (Figures 6 and 7).

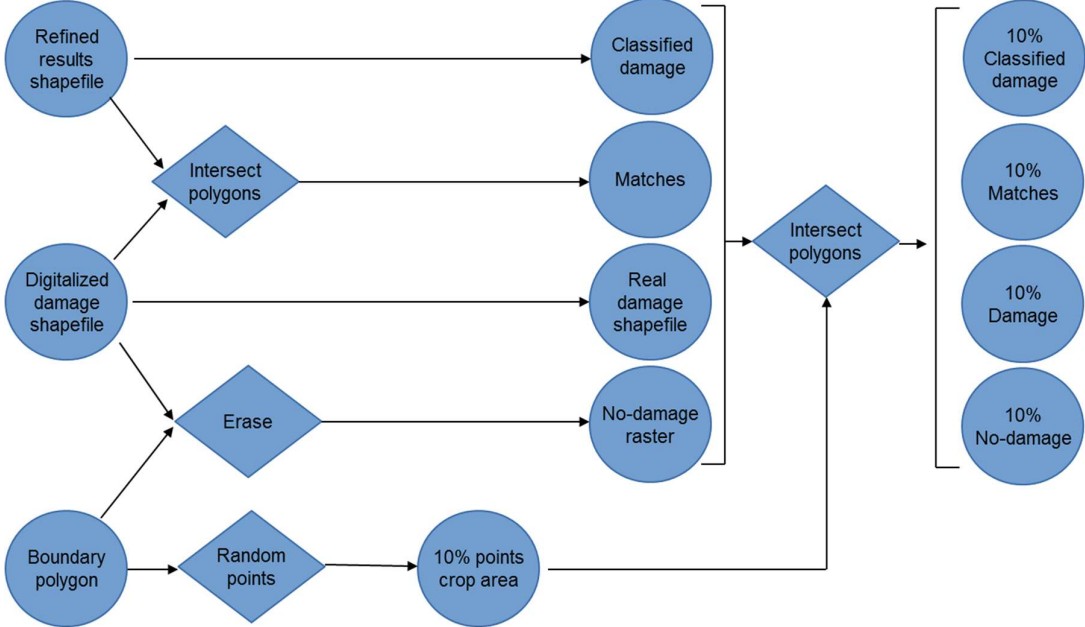

**Figure 6.** Workflow for the validation process of final results of 3D analysis.

The orthomosaics of the six study sites were used to manually digitalise polygons of damaged vegetation, following the expert criterion (Figure 8). The owners of the data provided the coordinates of the centroid of a damaged area as an example, which facilitated the photointepretation of the damage class.

For validation, two classes were defined: damaged and nondamaged vegetation. For each orthomosaic, a "damage" polygon vector was drawn by photointerpretation. By default, the "no-damage" class was the rest of the crop that is not damaged. A "no-damage" polygon vector was obtained by clipping a polygon vector that represented the boundary of the crop field with the "damage" polygon (Figure 8). These vectors were considered the ground truth data to validate the results of our study. The "damage" and "no damage" ground-truth data were crossed with the results of the crop damage workflow (from this point on, "classified" layer), to compare the percentage of coincidences and errors (Figure 8). By intersecting the "damage" vector with the results vector, two new layers were obtained: (1) a layer of the damage correctly classified, composed by the area that is coincident in both the ground-truth "damage" vector and the "results" layers; and (2) a layer that depict the errors, which is the area that was wrongly classified as "damage" in the results, which does not coincide with the photointerpreted "damage" vector (Figure 8).

According to the authors of [71], using a sample that represents the 15% of the total population for accuracy assessment leads to a pessimistic evaluation of the method. Therefore, in this study, a 10% of the surveyed area (whole crop field area) was used to evaluate the accuracy of the tested terrain analysis methods. To obtain a random sample from each crop field, the data (ground truth and results) were transformed from polygon vector to raster, and later into a point vector. The 10% of points were randomly selected. A distance between random points of 1 m was set (Figure 7).

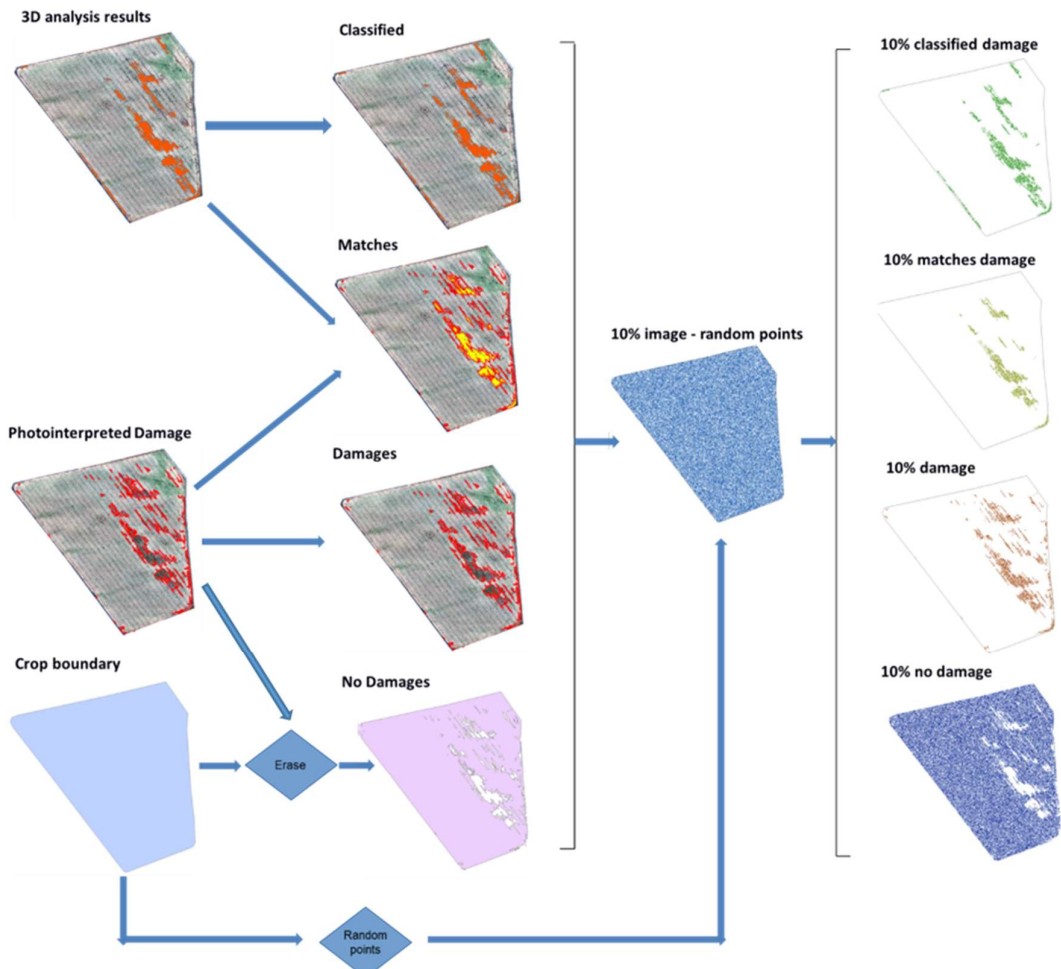

**Figure 7.** Validation workflow in images. Slope analysis on W1W as example.

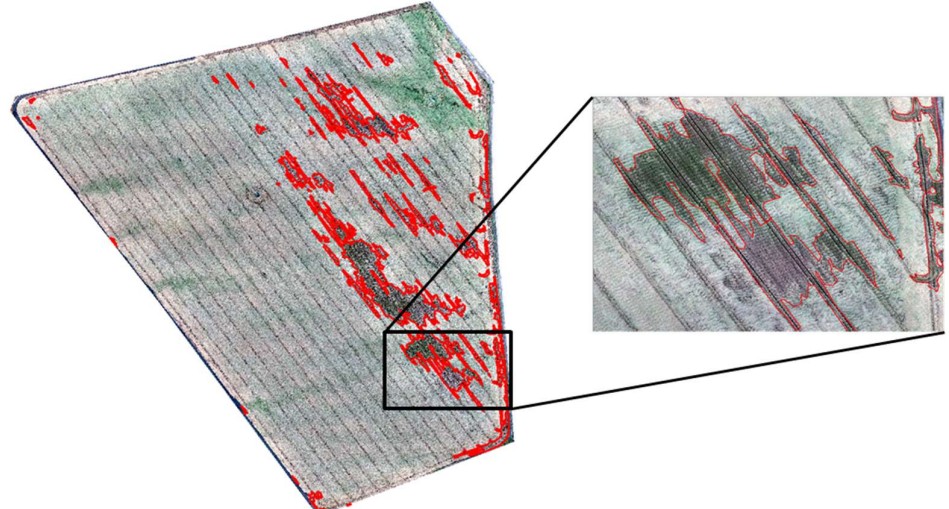

**Figure 8.** Photointerpretation and digitalisation of severe damages by expert observation. Slope analysis on study site "W1W" as example. See description of the site on Table 1.

The random points layer was intersected with the "damage" layer, the "no-damage" layer, the "matches" layer and the "classified" layer to obtain the 10% of random points of each class. The values of the class were transferred to the random points vector and these were used to calculate the accuracy of the methods.

The expressions used to evaluate the accuracy of the tested methods were defined by the authors of [72] (Figure 9):

$$Overall\ accuracy = (no\text{-}damage + matches) \times 100/n \tag{7}$$

$$Users'\ accuracy = 100 - ((damage - matches) \times 100/n) \tag{8}$$

$$Producers'\ accuracy = 100 - ((classified - matches) \times 100/n) \tag{9}$$

where:

*n*: # points at the 10% random points layer for the full surveyed area
*Damage*: # points at the 10% random points layer for the damage layer
*No-damage*: # points at the 10% random points layer for the no-damage layer
*Classified*: # points at the 10% random points layer for the classified layer
*Matches*: # points at the 10% random points layer for the matches layer

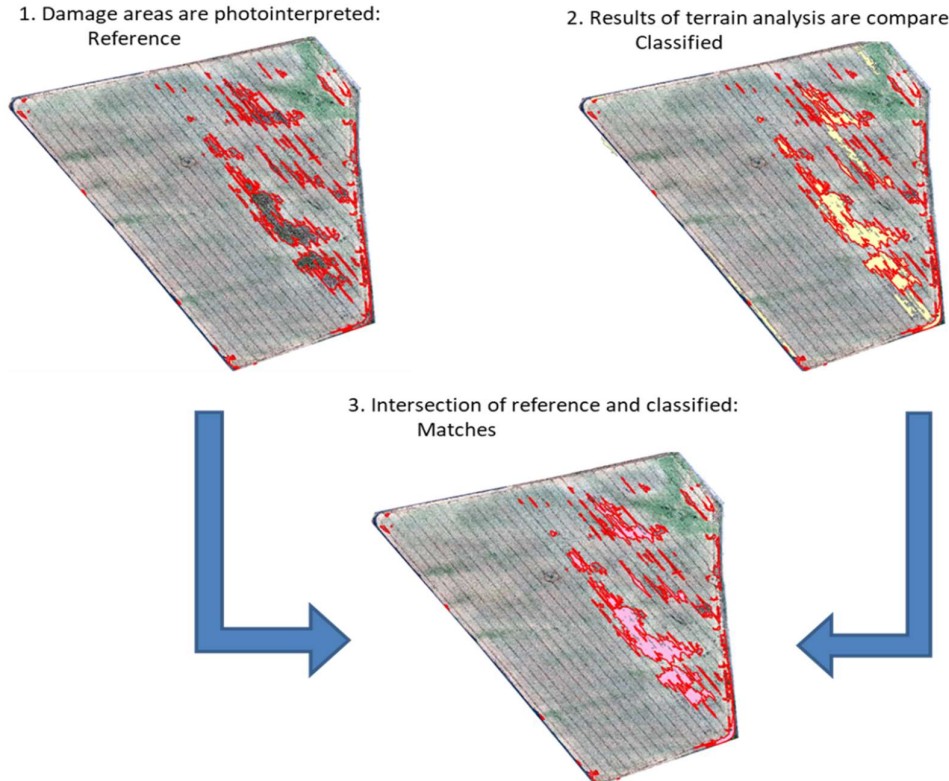

**Figure 9.** Calculation of matches by the intersection of photointerpreted damages (reference) and final results of the terrain analysis (classified). Slope analysis on study site "W1W" as example. See description of the site on Table 1.

## 3. Results

### 3.1. Calculation of Thresholds to Separate Damaged and Nondamaged Crops from Preliminary Results

As observed in the resulting rasters from Slope and Variance analyses, these methods depict a dramatic change of values along the edges of crop damages. In other words, the majority of values are low or high, with a low representation of intermediate values. A logistic function was used to separate the high from low values by finding the threshold value that separate edges of damage from the rest of values. These thresholds are shown in Table 4.

As observed in Table 4, the slope threshold delimiting the edges of damage is very similar in each DSM, which is very close to a value of 90 degrees, a vertical slope. On the contrary, the variance thresholds that define the edges of the damage are very variable from one to another DSM.

**Table 4.** Thresholds to discriminate edges of damage for slope and variance.

| Study Site | Slope | Variance |
| --- | --- | --- |
| | Slope Degrees | Decimal Degrees |
| B1L | 89.99 | 0.0005 |
| W1W | 89.994 | 0.1 |
| W2W | 89.99 | 0.01 |
| C1M | 89.9985 | 0.02 |
| C2A | 89.998 | 0.005 |
| C3W | 89.994 | 0.0025 |

In the case of CSF, we could not find a statistic to determine the optimal threshold to classify crop canopy depressions. In a preliminary exploration of the output data, it was observed that a classification threshold of 0.2 worked better than any other threshold for all DSMs. This parameter is related to the *xy* dimension, in other words, the pixel size. We set this as a fixed parameter. However, the cloth resolution parameter showed different results (*z* value) for the different tested crops, suggesting that the height of the crop (related to the crop type and stage of growth) influences the result of the tool. Namely, the height of the damage and the height of the standing vegetation are related.

*3.2. Crop Height Values Thresholds for Damaged and Nondamaged Vegetation*

The study cases present different crop heights, as they represent crops of different species and growth stage. Moreover, the intensity of the damage is different in every case and it is difficult to estimate the crop height in those cases. The farmers did not provide information about the mean crop height (nor in standing or damaged vegetation), as one of the purposes of the study was to develop a method to detect damages independently of characteristics of the crop fields.

Therefore, the height was assessed as the relative difference in crop height between standing and bent vegetation, independently of the absolute height values. The evaluation of the histogram allowed to separate damaged and nondamaged vegetation through a logistic function analysis [72]. Table 5 shows the crop height thresholds that allowed for the separation of healthy and damaged vegetation.

**Table 5.** Crop height thresholds to discriminate damaged from nondamaged polygons.

| Study Site | Slope | Variance | CSF | Geomorphology Classification |
| --- | --- | --- | --- | --- |
| B1L | −1.0000 | −2 | −20.0 | −8 |
| W1W | 100–1500 | 925 | 900.0 | 900 |
| W2W | 3 | Any except 0 | None | Any except 0 |
| C1M | 0–4000 | −5 | −40.0 | −125 |
| C2A | 0 | −3 | None | −5 |
| C3W | 15 | Any except 1 | None | 0 |

*3.3. Validation of Terrain Analyses*

In order to decide which method is more appropriate for the detection of crop damages in field crops, we calculated the overall accuracy and producers' accuracy for each study case (Table 6). The maps with all results are shown in the Supplementary Materials.

Since only two classes are explored (damage vs. no damage), "overall accuracy" equals "users' accuracy". According to the results, the four tested methods are highly accurate in the classification of crop damage, independently of the crop type, growth stage or damage type. Overall and users' accuracy are always between 87–99% accuracy, while producers' accuracy moves between 83–100%. The damage was underestimated in B1L, C1M (except for variance), C2A (except for CSF) and C3W (variance and CSF), and slightly overestimated for the rest of cases (Table 6).

**Table 6.** Accuracy assessment of the tested terrain analysis methods.

| Terrain Analysis Method | | B1L | W1W | W2W | C1M | C2A | C3W | Average |
|---|---|---|---|---|---|---|---|---|
| | Number of Samples | 189,900 | 90,500 | 11,900 | 7700 | 18,600 | 52,600 | |
| Slope | Overall accuracy | **98.9** | **96.6** | **97.3** | **97.0** | **95.7** | **95.8** | **96.9** |
| | User's accuracy | 98.9 | 96.6 | 97.3 | 97.0 | 95.7 | 95.7 | 96.9 |
| | Producer's accuracy | 97.6 | 97.7 | 80.9 | 97.6 | 83.1 | 95.7 | 92.1 |
| Variance | Overall accuracy | **98.9** | **98.0** | **88.6** | **96.2** | **92.6** | **95.8** | **95.0** |
| | User's accuracy | 98.9 | 98.0 | 88.5 | 96.2 | 92.6 | 95.8 | 95.0 |
| | Producer's accuracy | 97.2 | 96.5 | 89.2 | 97.2 | 84.9 | 94.1 | 93.2 |
| CSF | Overall accuracy | **98.0** | **97.0** | **95.5** | **96.3** | **94.4** | **92.6** | **95.6** |
| | User's accuracy | 98.0 | 96.9 | 95.5 | 96.3 | 94.4 | 92.6 | 95.6 |
| | Producer's accuracy | 99.1 | 94.4 | 90.0 | 99.1 | 94.9 | 98.1 | 95.9 |
| Geomorphology classification | Overall accuracy | **97.5** | **95.2** | **87.0** | **96.7** | **90.2** | **88.5** | **92.5** |
| | User's accuracy | 97.5 | 95.2 | 87.0 | 96.8 | 90.2 | 88.5 | 92.5 |
| | Producer's accuracy | 95.8 | 96.1 | 77.8 | 99.2 | 86.0 | 88.2 | 90.5 |

Based on users' accuracy, Slope and Variance achieved the best results, while the Geomorphology classification performed the worst, for all tested sites (Table 6). The accuracies vary from a 93–99% in Slope and Variance (with the exception of C1M site) to an 87–97% for Geomorphology classification (Table 6). According to the producers' accuracy, CSF provides the best results for most sites (90–99%), with the Geomorphology classification having the lowest values of the table, between 77–88% (Table 6). Therefore, Slope and Variance are good estimators of overall accuracy of crop damage, although slightly overestimating the damaged area, while CSF produces smaller errors of commission.

B1L presents the highest users' accuracy and performed well in producers' accuracy, together with W1W (Table 6). B1L corresponds to a barley cropland affected by logging, during the milking growth stage, in a terrain with heterogeneous topography (Table 1). W2W, C2A and C1M present the lowest users' and producers' accuracy (Table 6) for all methods, although still high, between an 87–97% users' accuracy and 77–90% producers' accuracy (Table 6). W2W is winter wheat damaged by wind, while C2A and C1M are corn fields damaged by wildlife and humans, respectively. In the three cases, the crops were in an advanced stage of growth (Table 7).

**Table 7.** Hectares and volume of damage and relative errors.

| Terrain Analysis Method | Study Site | B1L | W1W | W2W | C1M | C2A | C3W |
|---|---|---|---|---|---|---|---|
| | Reference Area (m$^2$) | 26,060.7 | 25,998.4 | 8037 | 2711.2 | 20,500.7 | 35,192.7 |
| Slope | Area (m$^2$) | 25,774.4 | 25,117.9 | 7818.9 | 2629.2 | 19,616.7 | 33,698.7 |
| | Error (%) | **1.1** | **3.4** | **2.7** | **3.0** | **4.3** | **4.2** |
| Variance | Area (m$^2$) | 22,770.9 | 25,483.3 | 7117.8 | 2608.7 | 18,990.7 | 33,703.4 |
| | Error (%) | **1.1** | **2.0** | **11.4** | **3.8** | **7.4** | **4.2** |
| CSF | Area (m$^2$) | 25,541.3 | 25,205.8 | 7674.3 | 2610.9 | 19,355.5 | 32,580.0 |
| | Error (%) | **2.0** | **3.0** | **4.5** | **3.7** | **5.6** | **7.4** |
| Geomorphology classification | Area (m$^2$) | 25,144.5 | 24,753.9 | 6995.6 | 2622.8 | 18,501.3 | 31,148.2 |
| | Error (%) | **2.5** | **4.8** | **13.0** | **3.3** | **9.8** | **11.5** |

For the selection of the most appropriate method, other factors apart from accuracy were considered, since all methods performed with an accuracy of over 90% (Table 6). The complexity of the process, data volume of intermediate and final files and computing time the workflow, using different terrain analysis methods, was calculated (Table 8). Slope was shown to be the method that consumes a smaller data volume and performs faster. On the other side of the ranking, Geomorphon was the most time consuming process and Variance was the process that produced a larger amount of intermediate data.

**Table 8.** Processing steps, data volume (value on the left, in Mb) and processing time (value on the right, in sec.) for each terrain analysis method. Values are average for all six study sites. (NA: Not Applied; NAcc: Not Accounted).

| Processing Steps | Slope | Variance | CSF | Geomorphon |
|---|---|---|---|---|
| **DSM Preprocessing** | | | | |
| DSM converted to point | 5.2 / 1.5 | 5.2 / 1.5 | 5.2 / 1.5 | 5.2 / 1.5 |
| Trend calculation | 0.6 / 3.9 | 0.6 / 3.9 | 0.6 / 3.9 | 0.6 / 3.9 |
| DSM detrending | 0.6 / 0.6 | 0.6 / 0.6 | 0.6 / 0.6 | 0.6 / 0.6 |
| DSM projection into UTM | NA | NA | 0.6 / 6.6 | NA |
| Conversion of re-projected UTM raster into points | NA | NA | 5.2 / 1.2 | NA |
| **Analysis** | | | | |
| Terrain analysis preliminary results | 0.2 / 0.3 | 78.4 / 1.0 | 0.2 / 0.2 | 0.06 / 15.0 |
| **Postprocessing** | | | | |
| Conversion of preliminary results into raster | NA | NA | 0.3 / 0.3 | NA |
| Generation of a mask for damaged pixels | 0.005 / 0.06 | 20.2 / 1.3 | NA / NA | 0.002 / 0.07 |
| Conversion of damage mask into polygons | 1.0 / 0.1 | 5.6 / 0.9 | 0.2 / 0.1 | 1.5 / 0.2 |
| Conversion of polygons containing mean elevation into raster | 0.8 / 0.3 | 0.8 / 0.6 | 1.0 / 0.3 | 0.8 / 0.4 |
| Conversion of CSM raster into integer values | 0.004 / 0.04 | 0.06 / 0.04 | 0.003 / 0.06 | 0.05 / 0.4 |
| CSM raster converted into polygons | 0.9 / 0.1 | 1.8 / 0.2 | 0.2 / 0.07 | 0.9 / 0.1 |
| Selection of polygons with lowest elevation | 0.3 / 2.3 | 0.03 / 2.3 | NA | 0.07 / 2.3 |
| Aggregation of polygons of same class | 0.03 / 0.3 | 0.03 / 0.6 | 0.003 / 2.1 | 0.003 / 0.3 |
| **Validation** | | | | |
| Photointerpretation of ground truth damage polygon layer | 0.001 / Nacc | 0.001 / NAcc | 0.001 / Nacc | 0.001 / Nacc |
| Generation of "No damage" layer | 0.04 / 0.08 | 0.04 / 0.08 | 0.04 / 0.08 | 0.04 / 0.08 |
| Generation of "Matches layer" | 0.2 / 0.2 | 0.003 / 0.6 | 0.004 / 0.1 | 0.4 / 0.3 |
| Generation of random points from CSM | 0.3 / 0.7 | 0.3 / 0.7 | 0.3 / 0.7 | 0.3 / 0.7 |
| Extraction of damage points from random points layer | 0.009 / 0.3 | 0.009 / 0.3 | 0.009 / 0.3 | 0.009 / 0.3 |
| Extraction of not damage points from random points layer | 0.5 / 0.8 | 0.5 / 0.8 | 0.5 / 0.8 | 0.5 / 0.8 |
| Extraction of classified points from random points layer | 0.009 / 0.3 | 0.004 / 1.3 | 0.006 / 0.4 | 0.004 / 1.1 |
| Extraction of matches points from random points layer | 0.006 / 0.3 | 0.05 / 0.3 | 0.006 / 0.3 | 0.006 / 0.3 |
| **TOTAL Data Volume (Mb)** | **10.6** | **114.2** | **15.0** | **11.0** |
| **TOTAL Computation time (sec)** | **11.9** | **16.7** | **21.6** | **27.6** |

In addition, a visual inspection of the results was done per terrain method, for the evaluation of the spatial distribution of errors. Figure 10 shows the results for one of the sites (C1M) across the four tested methods. To see all results for all study sites, please check the Supplementary Materials.

In the visual inspection, it was also observed that the edges of the damage were not so clear in W2W, C1M and C2A, which presented a lower accuracy for all terrain analysis methods. In these cases, the vegetation lays within the damaged areas, being the limits between healthy and damaged vegetation not as abrupt as in other cases. On the other hand, B1L, W1W and C3W—which present the highest crop damage detection accuracies—showed defined damages, with it being possible to see the ground in the damaged areas. Therefore, we can say that the efficiency of the methods depends on the severity of the damage.

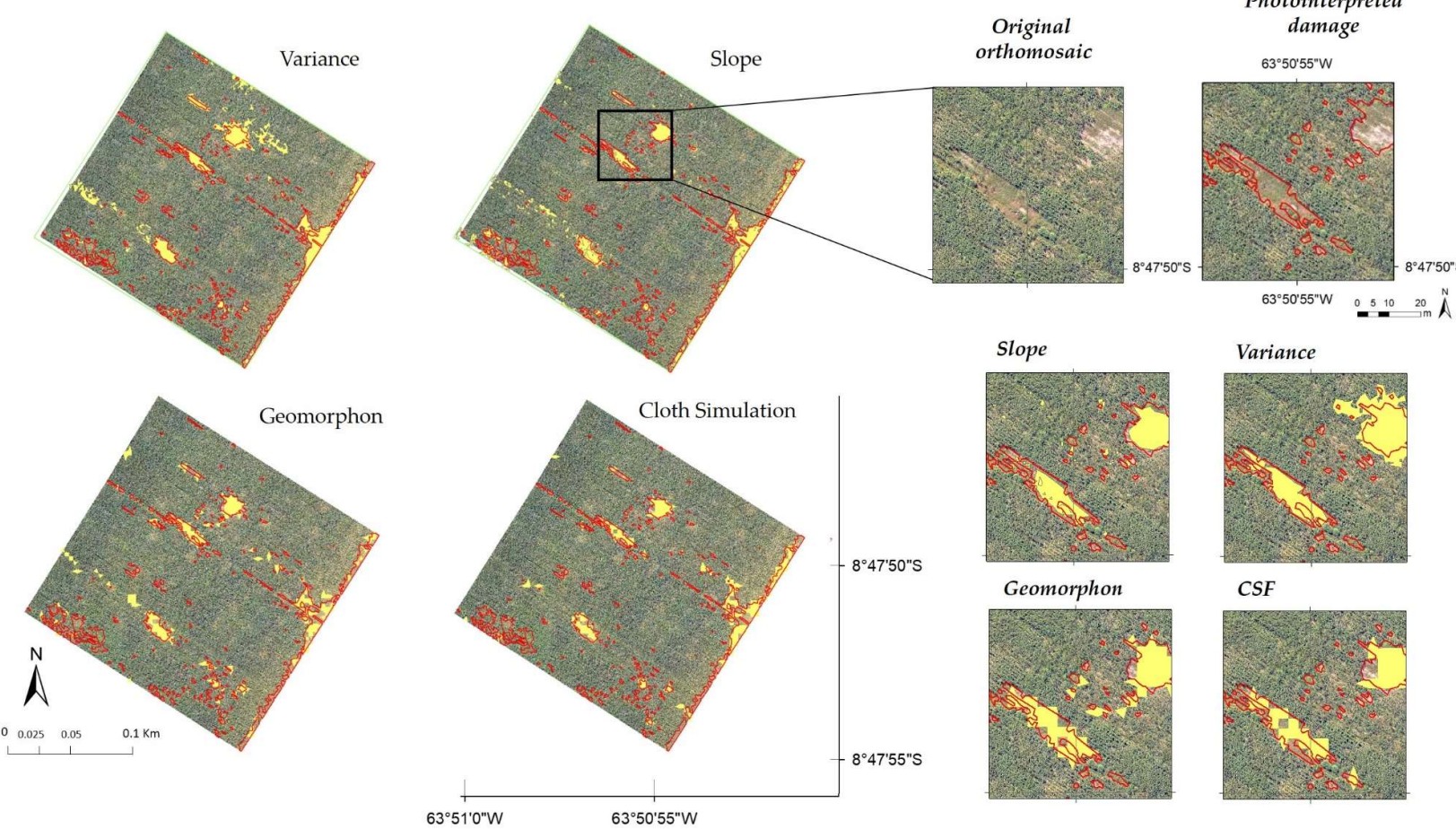

**Figure 10.** Final crop damage detection for C1M at the end of the workflow, using four different terrain analysis methods (Slope, Variance, Cloth Simulation Filter (CSF) and Geomorphology classification).



In addition, CSF was penalised in the selection of the best method because it required certain supervision from the user, since we did not find a way to calculate the classification threshold and cloth resolution parameters statistically from the data.

Considering that Slope was the fastest and simplest method (Table 8), the one that performed the best with an average overall accuracy of 96.9% among all tested sites (Table 6), and the method that presented the best spatial performance (Figure 10), this is the method that is best recommended for the detection of crop damages from UAV derived DSMs.

## 4. Discussion

Agriculture is the most important economic sector worldwide. At the same time, it is a risky activity due to many factors that are outside of human control or difficult to predict, such as weather and market fluctuations. The incorporation of remote sensing to agriculture management is largely improving the efficiency of the agrobusiness and facilitating farmers and other stakeholders' work.

Farmers and agronomists are incorporating UAVs into their daily crop management. However, most users use only some UAV functions, such as videos or single-shots of the entire cropland at a high elevation and low pixel resolution. Also, most users use basic RGB cameras and light low-cost UAS that do not allow sophisticated spectral analysis, but provide DSMs.

There are numerous studies that use DSM analysis for precision agriculture [45–47,49]; however, the use of DSM for crop damage estimation has been scarcely explored [29–32,40,73–76]. Michez et al. [77] proposed a method to detect damages in corn fields caused by wildlife, using UAV-based DSMs and height thresholds. In a similar fashion, Kuzelka and Surovy [78] used classification algorithms of crop heights in UAV-DSM to locate damage in wheat generated by wild boars, using accurate height estimations at the field taken with GNSS (Global Navigation Satellite System) for validating their results. Stanton et al. [79] reported that crop height extracted from SfM DSM and NDVI from UAV products were related to stress caused by aphid plagues. Similarly, Yang et al. [80] used a hybrid method of spectral and DSM data to classify logging damage in rice fields. As we see, the existing literature only evaluates study cases or use DSM data as a proxy. Our contribution to the sector is a workflow that can potentially be used in any case of structural crop damage, crop type and growth stage, in field crops that present significant damage intensity.

The goal of the present study is to provide a versatile and unsupervised tool, based on DSMs products and Geographic Information Systems analysis, for assessing crop damage after a disturbance. These events (i.e., wildlife, windstorms, fires) bend the vegetation or, in case of a very intense event, remove plant stands from the terrain. In effect, the damaged vegetation is relatively lower than the healthy vegetation and the damage is characterised as a depression or discontinuity in the crop canopy. These depressions can be modelled in Digital Surface Models (DSM) of the crop canopy and be used to detect damage in a terrain model analysis. The presented workflow delivers an estimation of damage in area units. Eventually, if the height of the crop is known, the damage can be estimated as a volume metric. If the yield and plant metrics from other years are known, yield loss can be inferred by regression analysis.

Since we did not find any terrain analysis developed specifically for agriculture, we used algorithms for other applications—such as hydrology, topography and forestry—for the analysis of crop canopies. Four terrain analysis methods were tested: Slope detection, Variance analysis, Geomorphology classification and Cloth Simulation Filter. A selection of six croplands located in different locations in America and Europe were used, representing different crop types (C: corn, W: wheat and B: barley), damage types (W: wind, A: animals, L: logging, M: Man-made) and growth stages, in order to evaluate the influence of those parameters in the accuracy of the tested terrain analysis methods. The results of this study revealed that all tested methods are very accurate detecting crop damage from UAV DSMs. The worst results are above the 77% (for Geomorphology classification), while some of the methods almost reached the 100% accuracy (98% for Slope and Variance analyses).

Our results also showed independence of our workflow to environmental and geographic factors, such as topography, soil type and irrigation management.

One of the challenges of this project was to create a tool that is able to analyze any generic DSM, independently of the UAV and camera types, in order to support a broad audience of farmers, agronomists and other agri-businesses' stakeholders. Also, we wanted our workflow to be independent of the observed target, covering large number of situations, such as different crop types, damage types and growth stage. From a technical point of view, one important achievement is the use of unsupervised methods. Other authors, such as Li et al. [41], used deep learning techniques to detect tree plantations in UAV images successfully. Deep learning overcomes problems such as projected shadows and colour differences in UAV imagery, at the cost of a large labelled database. A limitation of this study is that the workflow only works on field crops (i.e., wheat, barley, corn or canola). Damage in row crops would require different methods [74], which are based in image or 3D artificial intelligence algorithms [41]. The proposed workflow has been proved to be a versatile tool that delivers accurate crop damage estimations independently of the crop type, damage type and growth stage.

A major limitation of our study is that the presented workflow can only detect severe damage in crops. In this study, the damage intensity has not been specifically defined, among other reasons, because of the challenge of defining the term from a technical perspective. However, from the dataset it was possible to observe that the damage was more evident in some cases than in others. The orthomosaics allowed to observe that the workflow was more effective in those crops where the vegetation was completely bent than in the cases where only few plant stands were broken or partially bent. For instance, only Slope values above 89.9 degrees produced a good separation between damaged and not damaged vegetation. The same was observed for the rest of the methods: for Variance, only large variances in elevation; for Geomorphology tools, only large; and for CSF, only points in the clouds that were clearly separated from the surrounding points, were interpreted as damaged vegetation. That implies that damage at leaf level, such as that generated by hail, frost or diseases cannot be detected with this method. For damages that do not affect the plant structure, other methods based on spectral properties of the plant (i.e., using multispectral or hyperspectral sensors) can be useful [74,75]. However, rather than being a limitation, small damages are unlikely to cause losses in yield; therefore, the most significant damages (structural damages) are detected with our method.

A significant constraint of the workflow is the requirement for quality DSM as input data. Originally, around 20 farmers signed up as volunteers for this study. However, most DSMs had to be discarded because of low quality, which turned into artefacts that were interpreted wrongly as damage.

Since all terrain methods perform really well, the selection of the best terrain analysis was also evaluated taking into account technical factors, such as data volume, processing time, the number of processing steps and the possibility of a statistical selection of thresholds and parameters. The data volume generated was similar in all methods except for Variance, which was more than 3 times higher. CSF requires more steps for preprocessing and postprocessing; plus, it was not possible to select the appropriate parameters unsupervised. Therefore, Slope analysis was selected as the most accurate and efficient method for the detection of crop damage.

## 5. Conclusions

The present study intends to help farmers, agronomists and professionals of the agrobusiness in improving their field management and have accurate and objective estimations of damage to be used on insurance claims. In a world of a changing climate where it is expected to experience a rise in plagues, fires, floods, droughts and storms, unsupervised, accurate and fast estimations of crop damages will simplify the task of adjusters and farmers.

The current study explored existing terrain analysis methods to detect crop damage from Unmanned Aerial Vehicles (UAV) Digital Surface Models (DSMs). Datasets from different locations in Europe and America corresponding to different crop types, crop damages, growth stages and damage intensities were tested. This method was designed for field crops (i.e., barley, corn, wheat, rice, etc.)

and not for row crops. Four existing terrain analysis methods were tested: Slope detection, Variance analysis, Cloth Simulation Filter and Geomorphology classification. The proposed workflow did not require training data nor expert knowledge, but a *posteriori* refinement of the results was needed to remove machinery tracks and other sources of noise. The results of our study revealed that all methods are able to detect crop damage in our tested dataset with an accuracy above 90%, and that Slope and Variance were the methods that presented a higher overall accuracy, around 98%.

The presented workflow proved successful in terms of different crop type, growth stage, and damage type. However, our workflow is only effective for field crops data and severe intensity of damage.

Future research will focus on the implementation of the workflow into a programming language and the estimation of crop damage in biomass volume or yield by intersecting the information retrieved from this study (hectares of damaged crops) with information about the absolute height of the observed crops.

**Supplementary Materials:** The following are available online at http://www.mdpi.com/2072-4292/12/6/981/s1, Figure S1. Results of the six terrain analysis and post-processing workflow on dataset W1W (shown in yellow). In red, the damages detected from photointerpretation. UTM WGS84 Zone 34N, Figure S2. Results of the six terrain analysis and post-processing workflow on dataset C1M (shown in yellow). In red, the damages detected from photointerpretation. UTM WGS84 Zone 20S, Figure S3. Results of the six terrain analysis and post-processing workflow on dataset B1L (shown in yellow). In red, the damages detected from photointerpretation. UTM WGS84 Zone 14N, Figure S4. Results of the six terrain analysis and post-processing workflow on dataset W2W (shown in yellow). In red, the damages detected from photointerpretation. UTM WGS84 Zone 30N, Figure S5. Results of the six terrain analysis and post-processing workflow on dataset C2A (shown in yellow). In red, the damages detected from photointerpretation. UTM WGS84 Zone 33N, Figure S6. Results of the six terrain analysis and post-processing workflow on dataset C3W (shown in yellow). In red, the damages detected from photointerpretation. UTM WGS84 Zone 16N.

**Author Contributions:** V.E.G.M. contributed to the conceptualization, funding acquisition, data curation, methodology, validation, and writing (original draft and review and editing). C.R. contributed to the conceptualization, funding acquisition, methodology, project administration, resources, software provision, and supervision. G.A.S.-A. contributed to the funding acquisition, project administration, resources, software provision, supervision and writing (review). All authors have read and agreed to the published version of the manuscript.

**Funding:** This research was funded by the Mitacs Accelerate program (www.mitacs.ca/en/programs/accelerate) (Grant No 08301), with the collaboration of the University of Alberta and Skymatics Ltd.

**Conflicts of Interest:** The authors declare no conflict of interest.

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
