# Peer review of "Crop Loss Evaluation Using Digital Surface Models from Unmanned Aerial Vehicles Data"

_remotesensing, doi:10.3390/rs12060981_

Round 1
Reviewer 1 Report
The manuscript addresses the usage of different methods to estimate crop damage using photogrammetric outcomes from UAV-based data.
However, the manuscript needs to be improved to both meet the journal template (tables and references) and to improve the text, specially in materials and methods section.
Specific comments, suggestions and questions
Title
Remove “(DSM)” and “(UAV)”.
Abstract
Remove “(UAV)”, “(SfM)” and “(DSM)”, since these abbreviations were not used on the abstract.
Replace “this paper” by “this article” or “this study”, correct throughout the manuscript.
Rephrase one of the sentences that begin with “Our results”, to avoid repetition.
References
Please employ the reference style of this journal.
Introduction
Replace “drones” by “UAVs”.
Line 110: “This” instead of “The”?
Methods
Please, rename this section to “Materials and methods”.
Line 120: replace “irritation” by “irrigated”.
Table 1: Include the area of the study sites.
DroneDeploy should be described in the same way as the authors described ENVI 5.0 in line 236, instead of showing the URL of its website.
Section 2.2. does not provide relevant information.
It would be useful to provide some of the models or characteristics of the UAVs or RGB sensors used to acquired the imagery.
Format the tables according to the journal template.
Paragraph on line 180 has no connection to the previous text.
Did the authors considered to produce a crop surface model (CSM), by subtracting the DTM altitude to the DSM altitude? It is a relatively fast product to compute since both DSM and DTM can be computed using photogrammetric processing.
Last two paragraphs before Section 2.3.1 and Section 2.3.1. are confusing and should be rephrased to avoid having the same content spread through multiple sections.
There is no need to highlight the terrain analysis methods in Section 2.3.1. Moreover, the text in this section can be reduced.
Since most of the parameters shown in Table 3 are almost equal in all study areas this information could be included in the text.
Line 294: CSF uses point cloud data not DSMs correct accordingly throughout the manuscript.
Results
Add a new column to Table 6 with the mean value of the overall, User’s and Produccer’s accuracy for all study areas.
In the abstract it was stated that the methods performed nearly 100%, above or close to 95% would be more suitable.
Author Response
The manuscript addresses the usage of different methods to estimate crop damage using photogrammetric outcomes from UAV-based data.
However, the manuscript needs to be improved to both meet the journal template (tables and references) and to improve the text, specially in materials and methods section.
Answer: the manuscript has been largely changed, after a thorough review, after the recommendations of three reviewers and the Editor. The format for the journal has been considered and edited. We think the new version is much improved and will satisfy the expectations of the journal and the reviewers.
Specific comments, suggestions and questions
Title
Remove “(DSM)” and “(UAV)”.
Answer: Done.
Abstract
Remove “(UAV)”, “(SfM)” and “(DSM)”, since these abbreviations were not used on the abstract.
Replace “this paper” by “this article” or “this study”, correct throughout the manuscript.
Rephrase one of the sentences that begin with “Our results”, to avoid repetition.
Answer: All done.
References
Please employ the reference style of this journal.
Answer: Done.
Introduction
Replace “drones” by “UAVs”.
Line 110: “This” instead of “The”?
Answer: Done.
Methods
Please, rename this section to “Materials and methods”.
Answer: Done.
Line 120: replace “irritation” by “irrigated”.
Answer: Done.
Table 1: Include the area of the study sites.
Answer: Done.
DroneDeploy should be described in the same way as the authors described ENVI 5.0 in line 236, instead of showing the URL of its website.
Answer: We added the citation to the patent, but we would also like to leave the link.
Section 2.2. does not provide relevant information.
Answer: Removed.
It would be useful to provide some of the models or characteristics of the UAVs or RGB sensors used to acquired the imagery.
Answer: Unfortunately, that information was not recorded and the data is not in DroneDeploy anymore. At the time the data was downloaded I am not sure if DroneDeploy was even providing that information, neither. In any case, one of the points of the paper was to proof that the methods were independent of the cameras and only rely in the DSM. Just as a point, DroneDeploy only works with DJI, so all cameras should be similar. Only the model of the drone could change. My guess (and also, personal communication with some of the volunteer farmers) was that the most popular drone in the community was Phantom 2.
Format the tables according to the journal template.
Answer: Done by the editors.
Paragraph on line 180 has no connection to the previous text.
Answer: Right. We added an introductory sentence that connects that the pre-processing step of the workflow involves de-trending the DSM. Thanks.
Did the authors considered to produce a crop surface model (CSM), by subtracting the DTM altitude to the DSM altitude? It is a relatively fast product to compute since both DSM and DTM can be computed using photogrammetric processing.
Answer: Done. Somehow, that is what we are obtaining when de-trending the DSM. We could change the term DSM by CSM, if it is considered important. We used the term CSM many times in the new version of the manuscript. However, DSM is a more common term and more generic, so we would like to preserve it, at least in the title of the article.
Last two paragraphs before Section 2.3.1 and Section 2.3.1. are confusing and should be rephrased to avoid having the same content spread through multiple sections.
Answer: Done. Certainly, they were not well written. We re-wrote them, hoping to be better now.
There is no need to highlight the terrain analysis methods in Section 2.3.1. Moreover, the text in this section can be reduced.
Answer: Shortened.
Since most of the parameters shown in Table 3 are almost equal in all study areas this information could be included in the text.
Answer: Mentioned in the text. Do you imply that we should remove table 3? Please, let us know if so. Up to now, we left the table, but we will remove it if you find it necessary.
Line 294: CSF uses point cloud data not DSMs correct accordingly throughout the manuscript.
Answer: Done
Results
Add a new column to Table 6 with the mean value of the overall, User’s and Produccer’s accuracy for all study areas.
Answer: Done.
In the abstract it was stated that the methods performed nearly 100%, above or close to 95% would be more suitable.
Answer: Done.

Reviewer 2 Report
The article content is aligned with Remote Sensing journal. The authors used well known techniques to identify crop loss based on UAV DSM. As there is no methodological contribution, I suggest that the article become a letter or technical note.
Mainly a contextualization is presented in the introduction, and related works regarding the detection of crop modification are not presented. For this reason, it is not obvious the original contribution of the paper.
Observations regarding each section are presented below:
Abstract:
There is a confusion, because UAV DSM are not only generated using SfM but also with MVS method. Please, verify references about UAV Photogrammetry, and change this in all the text.
Please, review the following phrase: “Remote sensing data, image analysis and Structure from Motion (SfM) Digital Surface Models (DSM) offer new and fast methods to detect the needs of crops and are greatly improving crops efficiency”. Remote sensing data, image analysis and Structure from Motion (SfM) Digital Surface Models (DSM)
Instead of 100%, include the results per method.
Lines 20-21: It appears after the results. Please include the phrase “Other factors such as data volume, processing time and processing steps are discussed to select the most appropriate method”
- Introduction
In general, the authors only presented a contextualization in the introduction. I suggest to summarize this part, and include more details about related works regarding detection of crop modification. Only a paragraph was included to describe (lines 90-93).
Verify several grammar mistakes.
- Methods
Section 2.2 should be rewritten. The procedure to generate DSM based on UAV imagery is based on SfM and MVS. Please, include references to support the procedure description.
The reference polygons was delimitated using the DSM. Why the orthomosaic was not used? The edges in DSM is not always well defined.
- Results and discussion
I suggest to include a table with the following information: data volume, processing time and processing steps.
Instead of general results presented in appendix, I suggest to give a zoom in some area aiming at comparing the methods.
Is the appendix really necessary? I suggest to include some images in the results section. A zoom should be performed in some areas to present a qualitative comparison between the methods.
References
The references should be accompanied with number.
Author Response
The article content is aligned with Remote Sensing journal. The authors used well known techniques to identify crop loss based on UAV DSM. As there is no methodological contribution, I suggest that the article become a letter or technical note.
Answer: Thank you for your opinion. However, we disagree in the sense that not all papers contribute with a new methodological approach. Some papers, as the one proposed, test the validity of existing methods for certain applications. The novelty here is that we are using terrain analysis for a completely different approach: precision agriculture. Moreover, despite we could argue that the contribution of this study is, at least, moderate for the scientific community, it is of major interest for the industry, particularly the rising industry of commercial UAVs. Therefore, we do not see the incompatibility of publishing the content of this study in a scientific journal. We leave to the discretion of the Editor to decide if the content of the submitted manuscript is appropriate for the theme of the journal. We think it is.
Mainly a contextualization is presented in the introduction, and related works regarding the detection of crop modification are not presented. For this reason, it is not obvious the original contribution of the paper.
Answer: Again, we kindly disagree. In the introduction we show (1) studies that tested different methods for the detection of crop damage and other agriculture applications (lines 83-95 in the new manuscript, papers [54-57, 64-71], in the original manuscript); and (2) the motivation and objectives of our study.
If this is not enough, the discussion re-takes the topic of previous studies related to crop damage detection with remote sensing data and GIS techniques: lines 531 (papers 100-102); lines 534-546 (papers 54-57, 71, 103-106, in the original manuscript).
If we count all papers related only with crop damage detection with RS and GIS, there is a total of 12 papers cited in the manuscript, only for that topic. If we also count the papers related to other precision agriculture applications, with RS, then there are 7 more.
Observations regarding each section are presented below:
Answer: the manuscript has been largely changed, after a thorough review, after the recommendations of three reviewers and the Editor. The format for the journal has been considered and edited. We think the new version is much improved and will satisfy the expectations of the journal and the reviewers.
Abstract:
There is a confusion, because UAV DSM are not only generated using SfM but also with MVS method. Please, verify references about UAV Photogrammetry, and change this in all the text.
Answer: Good point, we also cited MVS. Thanks.
Please, review the following phrase: “Remote sensing data, image analysis and Structure from Motion (SfM) Digital Surface Models (DSM) offer new and fast methods to detect the needs of crops and are greatly improving crops efficiency”. Remote sensing data, image analysis and Structure from Motion (SfM) Digital Surface Models (DSM)
Answer: As said in a paragraph above, we modified the text largely. This line, too.
Instead of 100%, include the results per method.
Answer: in order to match with another reviewer, we changed to “all methods reached an accuracy above 90%” to be more accurate.
Lines 20-21: It appears after the results. Please include the phrase “Other factors such as data volume, processing time and processing steps are discussed to select the most appropriate method”
Answer: Done, thanks.
- Introduction
In general, the authors only presented a contextualization in the introduction. I suggest to summarize this part, and include more details about related works regarding detection of crop modification. Only a paragraph was included to describe (lines 90-93).
Verify several grammar mistakes.
Answer: We have shortened the introduction, especially the sections that are not directly related to the topic of the paper – the motivation of the study – and removed repetition. We expect that now it is lighter to the reader.
Please, also read our comments above.
- Methods
Section 2.2 should be rewritten. The procedure to generate DSM based on UAV imagery is based on SfM and MVS. Please, include references to support the procedure description.
Answer: It has been removed.
The reference polygons was delimitated using the DSM. Why the orthomosaic was not used? The edges in DSM is not always well defined.
Answer: the orthomosaics were used to validate the model, by the photointerpretation of the crop damaged areas. The main reason to not use the orthomosaic is that we are willing to provide a solution for commercial RGB cameras, which are largely used by farmers. Several studies (and our own experience) tell that the classification based on RGB data is not very successful. Most of the times, the classification must be supported with object segmentation to succeed. In our dataset, we identified problems of shadows of surrounding trees, poor light conditions during the flights, different patches of soil humidity, etc. We knew that there was a little chance to succeed in the classification of damaged vegetation by using RGB spectral data and suspected that we could succeed with the analysis of DSM (as we did!). The work with DSM avoided many problems related with “too simple” RGB spectral data. Our results prove that the edges of the damages are well defined in terms of DSMs, as long as the damage is severe, as stated in the paper. We added this point in the new manuscript.
- Results and discussion
I suggest to include a table with the following information: data volume, processing time and processing steps.
Answer: Done (current table 8)
Instead of general results presented in appendix, I suggest to give a zoom in some area aiming at comparing the methods.
Answer: Agree, we will move the appendix into Supplementary Material and we created a new figure (Figure 10) with one of the sites as an example, with zoom to an area of interest.
Is the appendix really necessary? I suggest to include some images in the results section. A zoom should be performed in some areas to present a qualitative comparison between the methods.
Answer: We agree. We do not like the appendix. The idea rose from a previous review; a reviewer asked for the results in maps. However, we noticed that we can submit those figures as “Supplementary Material” for mdpi Remote Sensing. We think that that can be a better option, together with a zoom example, as suggested before.
References
The references should be accompanied with number.
Answer: Right, we accommodated the references to the format of the journal. Thanks.

Reviewer 3 Report
The manuscript describes an unsupervised method to detect damages in cropland. The idea is promising and the proposed methodologies are interesting. However, the paper needs to be carefully revised by the authors. It is in fact too redundant with excessive references. Some sentences are suitable for a general-purpose book rather than a scientific paper concerning remote sensing techniques.
Concerning the general approach, the input dataset is highly biased with several unknow quantities and data. I strongly suggest repeating the whole process on a test site before attempting it on a "blind" one. Moreover, in some cases, it is difficult to understand why a certain parameter was selected. Equally, the recourse to expert knowledge is not so suitable with an attempt to develop an unsupervised procedure. Lastly, I recommend homogenizing the tool sources in order to make the procedure standardized and repeatable (a language programming rather the several GIS/software would be ideal).
Regarding the results, I would also suggest testing the possibility to integrate the four methods by combining them and cross-validating their results.
Author Response
The manuscript describes an unsupervised method to detect damages in cropland. The idea is promising and the proposed methodologies are interesting. However, the paper needs to be carefully revised by the authors. It is in fact too redundant with excessive references. Some sentences are suitable for a general-purpose book rather than a scientific paper concerning remote sensing techniques.
Answer: the manuscript has been largely changed, after a thorough review, after the recommendations of three reviewers and the Editor. The format for the journal has been considered and edited. We think the new version is much improved and will satisfy the expectations of the journal and the reviewers.
Concerning the general approach, the input dataset is highly biased with several unknow quantities and data.
Answer: Thank you. It would help us to know which quantities and data are unknown, according to the reviewer, in case we can clarify and rectify the text.
I strongly suggest repeating the whole process on a test site before attempting it on a "blind" one. Moreover, in some cases, it is difficult to understand why a certain parameter was selected. Equally, the recourse to expert knowledge is not so suitable with an attempt to develop an unsupervised procedure. Lastly, I recommend homogenizing the tool sources in order to make the procedure standardized and repeatable (a language programming rather the several GIS/software would be ideal).
Answer: The thing is that our personal challenge was to identify a technique (or more) that would work with any type of data, any type of structural damage and the less human component possible, with the smaller amount of information about the data. Basically, we wanted to take a DSM and work with it, independently of the source, UAV, camera, flying parameters, etc.
Our approach to “test a site”, as suggested by the reviewer, was the photointerpretation of damages on the orthomosaics and comparison to the results of the workflow, which is plenty accepted as a validation tool in the remote sensing community. Therefore, we did not test on one site, but six. We included the human supervision only to validate our data, but the methods themselves are unsupervised. It was not fully clear, though, so we modified the text to make it fully clear, that the selection of parameters and thresholds were calculated statistically, from the data and unsupervisedly.
About repeatability and coding, our method can be standardized and repeated. It is possible to write a code with the tested parameters and it will run successfully with any input data –however this was not our goal for this paper-. The slope, for instance, can be formulated in any programming language, it is simple trigonometry, by using the the formulas described in the manuscript. In the case of CSF or Geomorphon, their scripts can be directly called in a user- made script, since both methods have been written in R.
Maybe we were not clear in the manuscript and we try now to re-write things and include new paragraphs in the new version to highlight these points. Thank you for your opinion, it helps us identify points that were not clear.
Regarding the results, I would also suggest testing the possibility to integrate the four methods by combining them and cross-validating their results.
Answer: We kindly disagree. We do not see how the integration of four methods could improve the results, other than introducing noise in the models. The approaches are completely different and not complementary. In addition, we obtained a 98% of accuracy for slope, so there is little margin for improvement. How could the other methods help in refining the slope results, for instance?
About the cross-validation of results, after combining the four methods, we do not fully understand what the reviewer means. To our understanding, cross-validation is a technique that is used when the number of samples are small, so each sample is tested recursively against the rest of the population. However, this is not our case; we have plenty of samples (10% of the full dataset). Also, we do not understand how to do a cross-validation from the combined results of four methods. We would need further explanations to argue or accept this comment, sorry.

Round 2
Reviewer 1 Report
The authors improved the manuscript, by restructuring most sections and include the comments from the reviewers. However, some figures can be improved to not occupy such large space (e.g. Figures 2, 5, 9, 10) and to increase its quality, tables and the references Section are not yet according to journal template.
Author Response
Dear reviewer 1,
thank you very much for your time and your comments.
I contacted the Editor in charge of our manuscript and we are working on the format of tables. Also, I send the figures in JPG format, so the final quality of them will be appropriate in the journal view.
I will be in contact with the Editor regarding also the size of figures.
Thank you very much
Reviewer 2 Report
The authors improved the article, and present an interesting solution to detect trees using well known techniques applied to digital models generated using UAV imagery.
Observations:
Again, it was used only SfM in the introduction. As I mentioned in the round 1 revision (and the authors accepted), the DSM is generated using not only SfM but also MVS method.
In the introduction it was used “SfM or MVS”. UAV image processing software use both methods not only one. Please correct this.
3D or 3-dimensional are used. I suggest to use only 3D.
In the first review I suggested a revision in the introduction, because most of the section brings a contextualization. But the authors did not agree with my comments. But for example, many studies for diseases, plant nutrient is cited. Which is the relation of this with your paper? I suggest that you emphasize the approach. Unsupervised method is used in the abstract but it is not used in the introduction.
I agree that it is a difficult task to conduct tree detection in RGB images using traditional methods, for the reasons that you pointed out. But there are several papers, even published in Remote Sensing (2016-) that used Deep Learning (DL) for the detection of trees. DL overcome most of the reasons (shadow, etc). For example: https://www.mdpi.com/2072-4292/9/1/22. A disadvantage of DL methods is that is necessary a large labeled dataset. You can mention this, and show that your method did not require training dataset.
In my opinion, it is better to present papers that have some relation to yours, regarding tree detection instead of citing generic applications (diseased, nutrition, etc). It is not a review paper to have more than 100 references. I suggest to reduce it. You can cite a review paper instead of citing these generic papers.
Please correct reference [767]
Author Response
Dear reviewer,
thank you for your time and comments. We agree with them. We removed 24 references from the literature, mostly from the introduction section. If you check on the previous manuscript, these are:
3-7, 11, 16, 18, 19, 21, 22, 24, 34, 38-41, 66, 71, 72, 74, 94 and 96.
We removed some mentions to studies that are not related to our main scope (specially, those related to spectral analysis).
We corrected the mention to MSV (lines 13, 78 and 98).
We corrected the term "3D" (line 93).
We added the suggested paper about deep learning (lines 85 and 536).
We corrected reference (line 484).
Please, you can see the modifications in the new version of the manuscript, with track changes mode on.
Thank you!

Reviewer 3 Report
The authors have substantially changed, and improved, the paper. I'm still not convinced of the benefit of the extended explanation of the slope computation as it is a renowned procedure in GIS and related fields. I encourage the authors to remove or at least to reduce the section.
Author Response
Dear reviewer,
Thanks for your time and comments.
I double checked the section related to slope. Despite it is a very well known technique, I find it difficult and maybe not conveniente to reduce it, considering that it covers the same number of lines (or even less) than the other four methods. By removing the section or reducing it, it can be unbalanced in the overall methods section. Now, it is as:
Slope: 13 lines; variance: 13 lines; geomorphology classification: 15 lines, CSF: 18 lines.
Moreover, considering that it is the method that is selected (after our results and evaluations) as the most convenient method of all tested, it would be strange not to dedicate some lines to explain the method.
With all respects, we would like to keep it as it is.
Thank you!